# Gentamicin Population Pharmacokinetics in Pediatric Patients—A Prospective Study with Data Analysis Using the saemix Package in R

**DOI:** 10.3390/pharmaceutics13101596

**Published:** 2021-10-01

**Authors:** Paolo Paioni, Vera F. Jäggi, Romy Tilen, Michelle Seiler, Philipp Baumann, Dominic S. Bräm, Carole Jetzer, Robin T. U. Haid, Aljoscha N. Goetschi, Roland Goers, Daniel Müller, Diana Coman Schmid, Henriette E. Meyer zu Schwabedissen, Bernd Rinn, Christoph Berger, Stefanie D. Krämer

**Affiliations:** 1Division of Infectious Diseases and Hospital Epidemiology, University Children’s Hospital Zurich, Steinwiesstrasse 75, CH-8032 Zurich, Switzerland; vera.jaeggi@bluewin.ch (V.F.J.); romy.tilen@kispi.uzh.ch (R.T.); 2Biopharmacy, Department Pharmaceutical Sciences, University Basel, Klingelbergstrasse 50, CH-4056 Basel, Switzerland; roland.goers@unibas.ch (R.G.); h.meyerzuschwabedissen@unibas.ch (H.E.M.z.S.); 3Pediatric Emergency Department, University Children’s Hospital Zurich, Steinwiesstrasse 75, CH-8032 Zurich, Switzerland; michelle.seiler@kispi.uzh.ch; 4Department of Intensive Care and Neonatology, University Children’s Hospital Zurich, Steinwiesstrasse 75, CH-8032 Zurich, Switzerland; philipp.baumann@kispi.uzh.ch; 5Biopharmacy, Institute of Pharmaceutical Sciences, Department of Chemistry and Applied Biosciences, ETH Zurich, Vladimir-Prelog-Weg 4, CH-8093 Zurich, Switzerland; dominic.stefan.braem@alumni.ethz.ch (D.S.B.); cjetzer@student.ethz.ch (C.J.); robin.haid@bayer.com (R.T.U.H.); goetscha@student.ethz.ch (A.N.G.); 6Institute of Clinical Chemistry, University Hospital Zurich, Rämistr. 100, CH-8091 Zurich, Switzerland; Daniel.Mueller@usz.ch; 7Scientific IT Services, ETH Zurich, Binzmühlestrasse 130, CH-8092 Zurich, Switzerland; diana.coman@id.ethz.ch (D.C.S.); brinn@ethz.ch (B.R.); 8SIB Swiss Institute of Bioinformatics, Quartier Sorge-Batiment Amphipole, CH-1015 Lausanne, Switzerland

**Keywords:** dosing regimen, gentamicin, non-linear mixed-effects modeling, open-source, population pharmacokinetics, saemix, R-project

## Abstract

The aminoglycoside gentamicin is used for the empirical treatment of pediatric infections. It has a narrow therapeutic window. In this prospective study at University Children’s Hospital Zurich, Switzerland, we aimed to characterize the pharmacokinetics of gentamicin in pediatric patients and predict plasma concentrations at typical recommended doses. We recruited 109 patients aged from 1 day to 14 years, receiving gentamicin (7.5 mg/kg at age ≥ 7 d or 5 mg/kg). Plasma levels were determined 30 min, 4 h and 24 h after the infusion was stopped and then transferred, together with patient data, to the secure BioMedIT node Leonhard Med. Population pharmacokinetic modeling was performed with the open-source R package saemix on the *SwissPK*^cdw^ platform in Leonhard Med. Data followed a two-compartment model. Bodyweight, plasma creatinine and urea were identified as covariates for clearance, with bodyweight as a covariate for central and peripheral volumes of distribution. Simulations with 7.5 mg/kg revealed a 95% CI of 13.0–21.2 mg/L plasma concentration at 30 min after the stopping of a 30-min infusion. At 24 h, 95% of simulated plasma levels were <1.8 mg/L. Our study revealed that the recommended dosing is appropriate. It showed that population pharmacokinetic modeling using R provides high flexibility in a secure environment.

## 1. Introduction

The aminoglycoside antibiotic gentamicin is used in combination with a broad-spectrum β-lactam antibiotic as a first-line empirical treatment for suspected neonatal sepsis and other pediatric infections. As with other antibiotics, it is most effective if a minimal ratio between its plasma peak concentration and the minimal inhibitory concentration (MIC) of the target pathogen is achieved. However, in empirical treatment, the MIC is yet to be determined. The most common bacteria causing the observed infection are thus targeted. The pathogen causing the highest mortality rate due to sepsis in neonates is *Escherichia coli* [1]. Reported MICs that are effective against wild-type *E. coli*, i.e., without acquired drug resistance, range from 0.008 to 2 mg/L, with a maximum at 0.5 mg/L and 90% MIC values covered by 1 mg/L [2]. In adults, the recommended peak plasma levels of gentamicin are 8- to 10-fold that of the MIC of the target bacteria, resulting in plasma concentrations of 8–10 mg/L for an infection with typical wild-type *E. coli* [3,4]. Several other bacteria with relevance in neonatal sepsis, including *Staphylococci* and *Klebsiella* (wild-type strains), show a similar distribution of MIC to *E. coli* [2], making gentamicin a good choice for empirical treatment before the pathogen and its MIC are identified.

As is typical for aminoglycosides, gentamicin is potentially ototoxic and nephrotoxic and requires therapeutic drug monitoring (TDM). Several studies have revealed that toxicity is reduced by administering the daily dose all at once, allowing it to reach lower trough levels compared to dosing schemes with more-than-once daily administration [5]. The higher relevance of the trough than peak levels for toxicity is probably related to the fact that the mechanism of nephrotoxic accumulation in the tubular cells is saturable and thus concentration-independent when above the saturation concentration [5]. Consequently, current gentamicin monitoring focuses on the trough levels to ensure that plasma concentrations fall below 0.5 to 1 mg/L before the administration of the subsequent dose. Nevertheless, several authors cite 12 mg/L as the maximal plasma concentration, to avoid nephrotoxicity [3,6,7].

While trough levels of less than 0.5 or 1 mg/L can be achieved by adjusting the dosing interval, the choice of the peak levels and consequently of the dose means a compromise between drug efficacy and toxicity if bacteria with MIC > 0.5 mg/L need to be targeted. Touw et al. [3] considered 10–12 mg/L at 1 h after the infusion start of a 30-min infusion, and trough levels of between 0.5 and 1 mg/L as reasonable target concentrations for the effective but safe therapy of neonatal infections. Other studies reported peak levels of 5–10 mg/L as effective in neonates [3,8,9]. An analysis by van Donge et al. [10] demonstrated the challenge of choosing the right dose without knowing the precise MIC to target. Besides the trough and peak levels, the duration of the treatment is a further important factor for aminoglycoside toxicity. Peak levels above 12 mg/L may be tolerated for a few days but not for longer treatments [10,11,12,13].

The major challenge with predicting plasma levels and, eventually, dosing schemes for pediatric patients in general is that many influencing factors develop at their own pace between birth and adolescence. Differences in kidney function and body composition are assumed to be mainly responsible for inter-patient variability in gentamicin clearance and volume of distribution, respectively, in neonates and young infants [10]. Despite or as a result of the increasing understanding of gentamicin pharmacokinetics (PK) and the influencing factors, varying dosing recommendations exist in clinical care [13].

The aims of this prospective observational study were to (i) assess the PK of gentamicin in the studied patient population, (ii) identify covariates for predicting its PK, (iii) identify the dose appropriate to reach concentrations of between 10 and 12 mg/L at 30 min after the infusion stops, and predict the concentrations 24 h after the infusion start at the respective doses, and (iv) simulate plasma concentrations for typically recommended dosing schemes for the empirical treatment of suspected neonatal sepsis. Calculations were performed on the *SwissPK*^cdw^ platform within the secure environment of the BioMedIT node Leonhard Med [14,15,16], using the open-source software environment R (The R-Project for Statistical Computing) [17].

## 2. Materials and Methods

### 2.1. Patients and Data Collection

All patients aged from 1 d to 10 y, receiving gentamicin for at least 48 h at the University Children’s Hospital, Zurich (Switzerland), between 1 October 2017 and 30 April 2019, were eligible for enrolment in this prospective observational study. All study participants had a written informed consent signed by a legal representative, usually a parent. The study and research plan were approved by the Ethics Committee, Zurich, Switzerland (BASEC 2017-01296; date of approval, 6. September 2017); Appendix A). Patients with cystic fibrosis and patients receiving hemodialysis, peritoneal dialysis, or hemofiltration procedures were excluded. Two patients aged > 10 y (11.9 and 14.5 y) were inadvertently included. The total number of included patients was 109. Gentamicin was administered by infusion over 2 min or 30 min, depending on the usual practice in the corresponding unit, every 24 h, according to the hospital’s dosing recommendations. These were in general 5 mg/kg at an age <7 d after birth and 7.5 mg/kg at an age of 7 d or older. Blood samples for gentamicin quantification were taken immediately before an infusion (trough level), and ~30 min and ~4 h after the stopping of the second (or third) infusion. The gentamicin concentration in plasma was determined with the GEN/UniCel^®^ DxC 600 assay (Beckman Coulter, Brea, CA, USA) and for 5 patients (from 1 April 2019 onwards) with the Alinity C gentamicin assay (Abbott, Chicago, IL, USA). Concentrations measured with the GEN/DxC600 assay were transformed to agree with the Alinity C assay. The transformation function was determined with calibration samples. This corresponded to *C*(Alinity C, mg/L) = 0.832 × *C*(GEN/DxC600, mg/L) − 0.117 mg/L, with *C* as the gentamicin concentration.

### 2.2. Clinical Data Warehouse and Analysis Platform

Clinical data warehouses (CDW) are designed to store and harmonize clinical data for further use, e.g., to advance individualized medicine, including the optimization of dosing schemes. We have recently launched the Swiss Pharmacokinetics clinical data warehouse *SwissPK*^cdw^, in which we are able to collect, securely store and analyze clinical data from Swiss children’s hospitals [16]. The CDW contains routine clinical data as well as prospective study data. One of the goals of the CDW is to allow population pharmacokinetic modeling for optimizing the dosing recommendations for pediatric patients. Data confidentiality in accordance with the Swiss federal law is guaranteed by strategies developed within the Swiss Personalized Health Network (SPHN) [14,18]. All subjects are pseudonymized and the data are stored on Leonhard Med, a secure scientific platform for confidential research data [15]. Analysis of the data and modeling are performed directly within the secure environment of Leonhard Med [15]. The open-source programming language R [17] and the integrated development environment (IDE) RStudio [19] are available on Leonhard Med, including the dedicated R packages for non-linear mixed effects (NLME) modeling, *nlme* [20], *saemix* [21] and *nlmixr* [22]. The data of this study were analyzed within the secure environment of Leonhard Med.

Secure access to the data presented in this study is available on request according to the data sharing policies of SPHN [16,18]. The data are not publicly available for confidentiality reasons (Swiss Human Research Act, 810.30).

### 2.3. Data Analysis and Non-Linear Mixed Effects Modelling

Figure 1 provides a workflow of the data analysis. Data were analyzed within the open-source software environment R (3.5.1), using the IDE RStudio (desktop free version 1.1.456). We used the saemix (v 2.4) package for NLME modeling [21] and the packages *parallel* (v 3.5.1), *doParallel* (v 1.0.16) and *foreach* (v 1.5.1) for parallel computing. R scripts are deposited on gitlab.ethz.ch [23]. Comets et al. not only describe the saemix package in reference [21]; the publication also provides a good introduction to population pharmacokinetics, with examples.

Gentamicin plasma concentrations (*C*(*t*)) were fitted to an open two-compartment model with cumulative drug administration by infusion and linear (first-order) kinetics for distribution and elimination, according to Equations (1)–(5) [24]. Fit parameters were the apparent peripheral volume of distribution (*V*_2_′), the central volume of distribution (*V*_1_), the clearance (CL) and the apparent rate constant of distribution (*λ*_1_). They were used as their natural logarithm for fitting. The error model was “exponential” [21].
(1)$$\begin{array}{c} C\left(t\right)=\sum_{i=1}^{N}\frac{R_{0,i}}{{\mathrm{CL}}_{z}}\times \;[f_{1}\times \left(1-\exp\left(-\lambda_{1}\times t_{\inf,i}\right)\right)+\left(1-\exp\left(-\lambda_{1}\times T_{\inf,i}\right)\right)\times \exp\left(-\lambda_{1}\times t_{\mathrm{el},i}\right)\times \left(\left(T_{\inf,i}+T_{0,i}\right)<t\right)+\\ \left(1-f_{1}\right)\times \left(1-\exp\left(-\lambda_{z}\times t_{\inf,i}\right)\right)+\left(1-\exp\left(-\lambda_{z}\times T_{\inf,i}\right)\right)\times \exp\left(-\lambda_{z}\times t_{\mathrm{el},i}\right)\times \left(\left(T_{\inf,i}+T_{0,i}\right)<t\right)] \end{array}$$
(2)$$f_{1}=\frac{1}{V_{1}}\times \frac{k_{21}-\lambda_{1}}{\lambda_{z}-\lambda_{1}}\times \frac{\mathrm{CL}}{\lambda_{1}}$$
(3)$$k_{21}=\lambda_{1}\times \lambda_{z}\times V_{1}/\mathrm{CL}$$

In Equation (1), *N* is the total number of applied doses for an individual patient and *i* is the index of the individual dose. *R*_0,i_ is the rate of infusion of the *i*-th dose in mass/time. Parameter *λ*_z_ is the apparent elimination rate constant, calculated as CL/(*V*_1_ + *V*_2_′). The parameters *f*_1_ and *k*_21_ are described in Equations (2) and (3), with *k*_21_ the rate constant of redistribution from the second to the central compartment [24]. The start of the first infusion for an individual patient was set to *t* = 0. *T*_0,i_ is the starting time of the infusion of the *i*-th dose and *T*_inf,i_ is the duration of the infusion of the *i*-th dose. The time during the infusion of the *i*-th dose (*t*_inf,i_) and the time after the infusion stop of the *i*-th dose (*t*_el,i_) are relative to *T*_0,i_ and (*T*_0,i_ + *T*_inf,i_), respectively (Equations (4) and (5)). The logical terms in Equations (1), (4) and (5) correspond to 1 if they are true and 0 if false. The grouping factor for the NLME analysis was the subject pseudo-identity.
(4)$$t_{\inf}=\left(t-T_{0}\right)\times \left(\left(t-T_{0}\right)\leq T_{\inf}\right)\times \left(\left(t-T_{0}\right)\geq 0\right)$$
(5)$$t_{\mathrm{el}}=\left(t-T_{0}-T_{\inf}\right)\times \left(\left(t-T_{0}-T_{\inf}\right)\geq 0\right)$$

If CL was >(*λ*_1_ × *V*_1_) or *f*_1_ was <0 or >1 in the NLME analysis, the predicted subject *C*(*t*) would be set to 1000 in the objective function to introduce a penalty and, with that, omit these constellations as potential results. To avoid local minima of the objective function, the fits were run with multi-starts by parallel computing on 18 cores with *foreach* and *dopar* from the *foreach* R package, with random start values within defined ranges (Appendix A). The number of starts (random parameter sets) was 18. All parallel runs converged unless stated otherwise.

Correlations between the random effects of the NLME modeling and patient characteristics were analyzed by linear regression with the R function *lm*. Patient characteristics were expressed as the difference between their natural logarithm (except sex) and the natural logarithm of the reference value. The latter was close to the respective median value, as indicated. For missing values, this difference was set to 0, to avoid exclusion from the modeling. The number of subjects with available parameters is shown in Table 1. Patient characteristics that best correlated with the random effects (lowest *p* values from the linear regressions) were tested as covariates in the NLME modeling. Comparing two models (with different covariates), the model resulting in the lower −2 × log likelihood (−2LL) and lower Akaike information criterion (AIC) was considered significantly better if 1 minus the *p*-value for the chi-square distribution, with chi-square equal to the difference in −2LL and the degrees of freedom equal to the difference in the number of fit parameters, was lower than 0.05 (difference in −2LL > 3.84 for one additional covariate). At an equal number of degrees of freedom, the model with the lower −2LL and lower AIC was chosen. We considered the guidelines for reporting population PK analyses, published by Dykstra et al., in 2015 [25] and the EMA guideline from 2007 on reporting population PK analyses [26].

### 2.4. Treatment of Gentamicin Levels below the Limit of Quantification (LOQ)

Our data set contained 71 (out of 310) values below the LOQ (0.3 mg/L). Unless otherwise indicated, the residuals between fit and measured *C*(*t*) were set to zero if both the fit and measured values were below the LOQ. The residuals were modified accordingly within the objective function called by the *saemix* function, by setting the fit *C*(*t*) equal to the measured *C*(*t*) (both to LOQ) if both concentrations were below LOQ. In case the measured *C*(*t*) was below LOQ and the fit *C*(*t*) above, residuals equaled the difference between the fit *C*(*t*) and the LOQ. No modification occurred in the opposite case (fit *C*(*t*) below LOQ and measured above). We compared our fit parameters with those from fits where *C*(*t*) below LOQ were set to LOQ/2 without modifying the simulated values (as indicated). The two methods resulted in similar dose predictions and simulated plasma-concentration time curves (data not shown).

### 2.5. Validation of the Final Model

The final model was validated by computational and several graphical methods. The following plots were generated for graphical (visual) validation. (1) A scatterplot comparing the simulated (predicted) with the measured (observed) *C*(*t*) in linear and logarithmic scales. Simulations were on the population (simulations from the fitted fixed effects) and individual (simulations from the fitted fixed and random effects) levels. (2) Comparing the histograms of the observed plasma concentrations measured within 1 h after infusion stops and simulated *C*(*t*) at 30 min after infusion stops (*C*(30 min’)). (3) Residual random effects of the PK parameters plotted against the patient’s characteristics. (4) Residuals in *C*(*t*) compared with the simulated (predicted) *C*(*t*) and with *t*, respectively, both on population and individual level (ypred and ipred plots). (5) Histograms and Q-Q plots of the random effects. Q-Q plots were plotted with the *qqnorm* and *qqline* functions of the R package saemix. (6) Individual fit parameters plotted versus the −2LL values of the parallel runs of a multistart analysis. (7) Scatter- and line-plots of the observed and simulated *C*(*t*) against time, both for population and individual level and with *C*(*t*) in linear and logarithmic scale.

For computational validation, patient pseudo-identities were randomly re-sampled for bootstrapping, as indicated, with the R function *sample* (arguments *size* corresponding to the patient number, i.e., 109, and *replace* = TRUE to allow choosing a patient more than once), and analyses were run for parameter fitting according to the final model structure, with 18 parallel runs with random start values as described in Section 2.3 and Appendix A. A re-sampled data set contained on average 63% of the 109 individuals.

### 2.6. Prediction of the Dose to Reach a Defined Concentration 30 min after Infusion Stop

The dose (*D*) to reach a particular *C*(30 min’) was calculated according to a re-arranged Equation (1) (Equation (6)). Accumulation (*R*_ac_), i.e., the ratio between the *C*(30 min’) at steady-state and *C*(30 min’) after the first dose, was calculated according to Equation (7). The mean ± standard deviation of the individual calculated *R*_ac_ were 1.018 ± 0.027 and 1.028 ± 0.063 on population and individual levels, respectively.
(6)$$\begin{array}{c} D=C\left(30\;\min^{\prime }\right)\times \mathrm{CL}\times \frac{1}{R_{\mathrm{ac}}}\times \\ \frac{T_{\inf}}{f_{1}\times \left(1-\exp\left(-\lambda_{1}\times T_{\inf}\right)\right)\times \exp\left(-\lambda_{1}\times 0.5h\right)+(1-f_{1})\times \left(1-\exp\left(-\lambda_{z}\times T_{\inf}\right)\right)\times \exp\left(-\lambda_{z}\times (T_{\inf}+0.5h)\right)} \end{array}$$
(7)$$R_{\mathrm{ac}}=\frac{1}{f_{1}\times \left(1-e^{-\lambda_{1}\times \tau }\right)+\left(1-f_{1}\right)\times \left(1-e^{-\lambda_{z}\times \tau }\right)}$$

### 2.7. Simulations of Plasma Concentrations with Defined Doses

To simulate the concentration ranges for the studied population for different doses, we calculated 100 concentration-time curves for each individual, using Equation (1). The PK parameters for these simulations corresponded to the subject’s parameters at population level plus for each parameter a random value according to *Ν*(*µ*,*σ*^2^) with *µ* = 0 and *σ* equal the standard deviation of the respective random effect. The 95% and 99% confidence intervals (CI) of the simulated concentration-time curves were defined by clipping the minimal and maximal 2.5% and 0.5% simulated concentrations, respectively, at each time point.

## 3. Results

### 3.1. Population Description

The characteristics of the patients are summarized in Table 1. Appendix A show the distribution of the patient characteristics and their correlations, as well as the distribution of the once-daily dose in mg per kg body weight. The primary diagnoses and their frequency are shown in Table 2. The total number of available gentamicin plasma concentrations was 310, including 71 measurements below the limit of quantification (LOQ). The LOQ was 0.3 mg/L.

Up to three gentamicin plasma concentration measurements were available per dosing interval. Figure 2 shows all measured concentrations plotted against time passing after the start of the first infusion. Appendix A shows the logarithmic concentrations versus time for the individual subjects. For 24 of the 109 subjects, 3 plasma concentrations above the LOQ (0.3 mg/L) were available within one dosing interval (Appendix A). The characteristics of these patients exhibited similar distributions as the complete study population (Appendix A and Table 2).

### 3.2. Structural Model Description

For 22 of the 24 subjects with 3 measured plasma concentrations above the LOQ within one dosing interval, the second plasma concentration was lower than expected for a one-compartment model (Appendix A), in favor of a two-compartment model. We, therefore, analyzed the data with an open two-compartment model with cumulating dosing, zero-order infusion and first-order distribution and elimination, including all 109 patients and all available concentrations. Fit parameters were ln(CL), ln(*V*_2_’), ln(*λ*_1_) and ln(*V*_1_), according to Equation (1). Random effects were included for ln(CL), ln(*V*_2_′) and ln(*V*_1_), but not for ln(*λ*_1_), to avoid overparameterization. We considered *λ*_1_ as the PK parameter that is least dependent on patient characteristics. For a final evaluation, the inclusion of random effects for ln(*λ*_1_) was tested with the final model. The patient characteristics shown in Table 1 and Appendix A were evaluated as potential covariates for the fit parameters, as described in the next section (Section 3.3. Covariate Analysis).

### 3.3. Covariate Analysis

To identify potential covariates, the function in Equation (1) was fit to the data in the absence of covariates, and the random effects of the fit fixed effects ln(CL), ln(*V*_2′_) and ln(*V*_1_) were plotted against the available patient characteristics. The potential covariates were expressed as the difference between their ln and the ln of a reference value (close to the respective median, Table 3), except for sex, which was set to 0 for male and 1 for female. The correlations are shown in Appendix A.

The strongest correlation was identified between the random effects of ln(CL) and the ln(bodyweight), with an adjusted *r*^2^ = 0.586 and *p* < 1^−6^ (Appendix A). The correlation was similarly strong for ln(body surface area; adjusted *r*^2^ = 0.576, *p* < 1^−6^). We included ln(bodyweight) as a covariate for ln(CL) and did not test ln(body surface area) as an alternative, as it is a calculated parameter (Appendix A). Including additional covariates that correlated with the random effects improved the model further. The improvements were significant (−2LL reduction by >3.84 for an individual covariate) including ln(body weight) as a covariate for both ln(*V*_2_′) and ln(*V*_1_), and including ln(creatinine concentration) and ln(urea concentration) as further covariates for ln(CL). Leaving one covariate out significantly increased −2LL. Including sex as a covariate for ln(CL) reduced −2LL, but by less than 3.84. Sex was, therefore, not included as a covariate in the final model. Appendix A shows the comparisons of the patient characteristics and the remaining random effects of the final model.

Finally, we tested the inclusion of random effects for ln(*λ*_1_) in the final model. The value of −2LL for the best fit (lowest −2LL of the 18 parallel runs) decreased by 60, which would be in favor of including the random effects of ln(*λ*_1_) in the final model. However, of the 18 parallel runs with random start values, only 12 converged to meaningful results. In addition, the ln(*λ*_1_) random effects did not suggest any covariates for ln(*λ*_1_); the adjusted *r*^2^ of the correlations between the random effects of ln(*λ*_1_) and the patient characteristics were all < 0.024. As the final fit values did not substantially differ, whether the random effects of ln(*λ*_1_) were included in the model or not (data not shown), we did not include them in the final model, in order to keep the model robust.

### 3.4. Final Model

Table 3 shows the structure and fit parameters of the final model. The ranges of the fitted PK parameters for the studied patient population are shown in Table 4. Appendix A shows the CL in mL/min/1.73 m^2^, plotted against age and compared to reference values for the glomerular filtration rate (GFR; [27]). The fit CL (individual levels) was scattered between ~50% and ~100% of the age-dependent reference values for GFR.

### 3.5. Validation of the Final Model

In Figure 3, we provide the comparisons between the simulated (predicted) and the measured (observed) *C*(*t*) at population and individual levels for the final model. Regarding the population level, 98 *C*(*t*) values were predicted as ≤1 mg/L, 93 of them were also measured as *≤*1 mg/L. The five *C*(*t*) with a simulation ≤ 1 mg/L but measurement > 1 mg/L were from 4 male patients and 1 female patient, and included the 2 patients of the study with gestational age < 37 weeks and age < 7 days (see Table 1) and one patient with serum creatinine > 50 µM (Figure 3c). Of the 99 measured as *C*(*t*) ≤ 1 mg/L, 93 were also predicted as <1 mg/L (Figure 3c).

Appendix A show the residuals compared to the simulated (predicted) *C*(*t*) or compared to time, both on the population and individual levels. We did not observe a major asymmetry of the residuals with respect to their 0-lines. The histograms and Q-Q plots (sample quantiles compared to theoretical quantiles) of the random effects are shown in Appendix A. Of the three PK parameters with random effects (ln(*V*_2_′), ln(CL), ln(*V*_1_)), the random effects of ln(CL) were closest to a normal distribution, while both the histograms and Q-Q plots of the logarithmic volume terms indicated some tailing on both sides.

Appendix A shows the fitted fixed effects of all runs of a multistart calculation, with 18 sets of start values, randomly generated within the ranges in Appendix A. For the 10 lowest −2LL values, fit parameters deviated less than 0.09 (ln scale) from the respective parameter at the lowest −2LL. Including all 18 runs, the highest deviation was 0.16. The range of −2LL was 2.1. The plot indicates that the multistart approach, with 18 sets of random start values, provided robust and reproducible results when applying the final model structure. For further inspection, Appendix A compare the observed *C*(*t*) and ln(*C*(*t*)) with the simulated *C*(*t*) and ln(*C*(*t*)), respectively, of the individual subjects versus time, on population and individual level.

Besides these visual evaluations, we validated the model by bootstrapping with 1000 re-sampled data sets. The random data sets were analyzed using the final model structure. Appendix A shows the histograms of the individual fit parameters of the bootstrapping, with their means and 95% CI (assuming normal distribution), in comparison with the fit results from Table 3. Fit parameters from Table 3 were within 95% CI of the bootstrapping results. Conversely, the means of the bootstrapping results were within 95% CI of the fit parameters from Table 3 (Appendix A). Except for ln(*V*_1_) and ln(*λ*_1_), the CI of bootstrapping and fit parameters in Table 3 were similar. The parameter histograms from bootstrapping for ln(*V*_1_) and ln(*λ*_1_) were bimodal (Appendix A), resulting in a broadening of the CI compared to the fit parameters in Table 3. These data show that while most fit parameters were robust towards re-sampling, fitted ln(*V*_1_) and ln(*λ*_1_) depend on the re-sampled population. The narrower CI of the fit parameters in Table 3, as well as the constant fit parameters in Appendix A, indicate that the results from the complete population were robust. The bootstrapping results reflect the sparsity of individuals with 3 concentration measurements above the LOQ within one dosing interval (*N* = 24). The model may become over-parametrized in the absence of such subjects from the re-sampled population.

Bootstrapping from 100 re-sampled data sets with the final model, including random effects for ln(*λ*_1_), resulted in a high number of outliers in the fit parameters and thus, a broad CI over several ln units, supporting the decision to omit random effects for ln(*λ*_1_) in the final model (data not shown).

### 3.6. Dose Prediction to Reach C(30 min’) between 10 and 12 mg/L

The concentration at 24 h after dosing (*C*(24 h’)) was available for 87 of the 109 patients. Of these, 5 had *C*(24 h’) > 1 mg/L, one of them > 2 mg/L (2.5 mg/L). Regarding the *C*(30 min’), 83 patients had a concentration measurement within 1 h after infusion stops. Of these, 73 had at least one measured gentamicin plasma concentration above the recommended maximal level for long-duration treatment of 12 mg/L. Figure 4 shows the distributions of the simulated *C*(30 min’) at the population and individual levels, compared to the measured *C*(30 min’).

We used Equation (6) with the fixed effects of our final model (Table 3) to predict the once-daily dose to reach *C*(30 min’) = 10.95 mg/L, i.e., the geometric mean of 10 and 12 mg/L. As the dose should be easily assessable with a routinely determined parameter, we compared the predicted dose with age, body weight and body surface area (Appendix A). As expected from the model, the correlation was best fitted with body weight, the covariate for the volume terms and CL. Figure 5 shows the predicted dose for 2- and 30-min infusion time in relation to body weight and compared to the given dose.

We used the linear relationship according to Equation (8), between the predicted individual doses (*D*) and body weight (Figure 5), as a general dose suggestion for our study population to reach *C*(30 min’) of between 10 and 12 mg/L.
*D* (mg) = *c* (mg) + *d* (mg/kg) × BW (kg)(8)

In Equation (8), *c* is the intercept (constant value in mg) and *d* is the slope in mg/kg between predicted dose and body weight (BW). Simplified from the linear regression in Figure 5, we simulated plasma concentrations with *c* = 6 mg and *d* = 4 mg/kg (Equation (8)), using the final model. Plasma-concentration time curves were simulated for each individual, with the respective population fixed effects and simulated random effects, as described in the Methods section. In Figure 6, the resulting simulations for the plasma-concentration time curves over 3 dosing intervals with 30 min infusions, and the respective distributions of the *C*(30 min’) and C(24 h’), are given. For the respective data for 2-min infusions, we are referring to Appendix A.

### 3.7. Simulations of Plasma Concentrations with Typical Once-Daily Doses

There is no one-fits-all dosage recommendation for gentamicin, as the desired target peak concentration or *C*(30 min’) depends on the target MIC, taking into consideration the duration of the treatment and the tolerated risk of toxicity. We, therefore, simulated plasma concentrations, with typical recommended once-daily doses in addition. These were 3 mg/kg (not typically recommended), 4 mg/kg, 5 mg/kg and 7.5 mg/kg. Plasma-concentration time curves were simulated as described in the former section. The simulations are shown in Figure 7 and Appendix A. As expected, the 2-min infusion time revealed higher *C*(30 min’) than the 30-min infusion time. At 3 mg/kg once daily, 95% of the simulated *C*(30 min’) were between 5.2 and 8.4 mg/L (30 min infusion) or 5.8 and 10.0 mg/L (2 min infusion time). At the highest dose (7.5 mg/kg once daily), the respective ranges were 13.0 to 21.2 mg/L and 14.6 to 25.0 mg/L.

## 4. Discussion

Gentamicin dosing in neonates and infants remains a debated field, despite the numerous population PK analyses, recently reviewed by Llanos-Paez et al. and Crcek et al. [13,28]. As discussed by van Donge et al., short durations of empirical treatment at relatively high doses may be tolerated, while peak levels may be more critical for longer treatment durations [10]. After a short duration of empirical treatment at a relatively high dose, the dose needs to be adjusted once the MIC is available. Taking these considerations into account, our simulations aimed to predict the plasma concentrations of gentamicin in our study population at typical once-daily doses rather than suggesting an optimized dose for all purposes.

In agreement with the aim of empirical treatment, to target bacteria with potentially high MIC, most patients in our population reached higher *C*(30 min’) than the commonly suggested maximal value of 12 mg/L for long-term treatment. The most frequently administered dose was 7.5 mg/kg, once daily. The respective simulated *C*(30 min’) with a 30 min infusion time were within a 99% CI of 12.4 to 25.4 mg/L. Assuming efficacy at *C*(30 min’) ≥ 8-fold the MIC [3], a once-daily dose of 7.5 mg/kg would be effective against bacteria with MIC = 1.5 mg/L in > 99% of the pediatric patients in our study population. Efficacy against bacteria with a MIC of 1 mg/L would be achieved at a 5 mg/kg once-daily dose (30-min infusion), as 99% of the simulated *C*(30 min’) were between 8.2 and 16.9 mg/L.

Importantly, according to our simulations, *C*(30 min’) ≤ 12 mg/L would be achieved in 95% of patients at a once-daily dose of 4 mg/kg, administered in a 30-min infusion (6.9 to 11.3 mg/L). This once-daily dose would agree with the recommendations for the age group ranging from 7 days to 1 month in the Dutch Pediatric Formulary (DPF). Hartman et al. [29] recently reported data from a retrospective study in which most included patients reached peak concentrations of < 8 mg/L with the Dutch dosing scheme (infusion duration and time point after infusion stop not further defined). Wang et al. [30] predicted a dose of 4 mg/kg at birth to target a peak level of 10 mg/L. Their predicted dose decreased to 3 mg/kg at 2 years of age. Considering the abovementioned results by the authors and others, the 4 mg/kg once-daily dose appears to be the lower limit of empirical dosing and to be rather appropriate after a MIC of < 1 mg/L was validated.

According to our simulations, *C*(30 min’) only marginally depends on the duration of the infusion. *C*(30 min’) was 10–20% higher for the 2-min than the 30-min infusion, in agreement with the measured concentrations. For the simulated doses up to 4 mg/kg as a once-daily dose, the simulated *C*(24 h’) were below the recommended maximal concentration of 1 mg/L in ≥95% of the patients in our study (both 2 and 30 min infusion), in agreement with other studies [11,13]. However, our findings need to be treated with caution as we did not include patients with impaired kidney function, and our study involved only 2 patients at a gestational age < 37 weeks, combined with age after birth < 7 days. Their measured *C*(24 h’) was 1.2 and 1.4 mg/L, respectively. Prolongation of the dosing interval is recommended for these patients [10,28].

In our study, we used CL, *V*_1_, *V*_2′_ and *λ*_1_ as fit parameters for the two-compartment model. As in other groups before, we identified body weight and serum creatinine as significant covariates for the CL [13,28]. In our study, the serum urea concentration was a further significant covariate for CL, although it is commonly described as a poor predictor of GFR [31]. It may have compensated for the lacking serum creatinine values of <27 µM. The CL determined for our study population (individual level) was between 1.5 and 71.0 mL/min (median CL normalized to 70 kg body weight was 7.0 L/h/70 kg), in agreement with the CL determined by Germovsek et al. (6.2 L/h/70 kg at 0.5 to 5.1 kg body weight) [32]. Our surface-area normalized range of CL (8.7 to 138 mL/min/1.73 m^2^, individual level) corresponded to between ~50% and ~100% of the reported GFR for the respective age groups [27]. This is in agreement with filtration of the unbound drug, where the free fraction in plasma is >0.7 [33], with negligible re-absorption due to the high hydrophilicity of the drug.

The fit *V*_1_ and *V*_z_ indicate the initial distribution of gentamicin in the extracellular body water (median *V*_1_ on an individual level was 20.6 L/70 kg) with steady-state distribution in the total body water (median *V*_z_ on individual level 37.1 L/70 kg). Our modeled *V_z_* was in the range of the *V*_z_ of other studies, employing a two-compartment model [13]. The identified covariate bodyweight for *V*_1_ and *V*_z_ is also the one most frequently used in other studies [13,28]. We used *λ*_1_ to describe the kinetics of the distribution phase. Due to the scarcity of the data, we did not compute random effects for this parameter.

We only tested linear relationships between the (logarithmic) PK parameters and the (logarithmic) patient characteristics, while other studies used more complex relationships [13,28,30]. Linear relationships are typical for studies without a priori knowledge [13]. Neither the initial nor the remaining random effects in our final model indicated any strong non-linear relationship.

Modeling with R, using the saemix package, allowed us to treat gentamicin concentrations below the LOQ in an unconventional way. Any method of handling below LOQ values introduces a bias. Discarding the values below the LOQ will result in an overestimation of the predicted value for the respective time point and, thus, the underestimation of CL. Setting them to LOQ without further modification has a similar effect. Setting them to zero underestimates the average concentration and overestimates the CL. As a compromise, below-LOQ values are often set to LOQ/2. In our method, we set the residue to zero if both the predicted and measured values were below LOQ. The only bias occurred when the predicted value was above the LOQ and the measured was below, as the true residue would be higher than the chosen difference between the predicted value and LOQ. However, bringing the predicted value below the LOQ (same as the measured value), reduced the residue (to zero) in any case. Our suggested method revealed similar results when setting below-LOQ values to LOQ/2. We will continue evaluating our proposed method in future studies.

## 5. Conclusions

We confirmed that a dosing scheme with once-daily 7.5 mg/kg at age 7 d or older and 5 mg/kg at age < 7 d reaches plasma concentrations for targeting bacteria, such as *E. coli*, with relatively high MIC (≥1 mg/L) and is, therefore, suited for the short-term empirical treatment of suspected neonatal sepsis. Our simulations at various typical doses can guide dose-finding once the MIC is determined. Modeling with the saemix package in R allows high flexibility in data treatment and analysis. With this study, we successfully test-ran the *SwissPK*^cdw^ platform by population PK modeling using the open-source software language R exclusively.

## Figures and Tables

**Figure 1 pharmaceutics-13-01596-f001:**
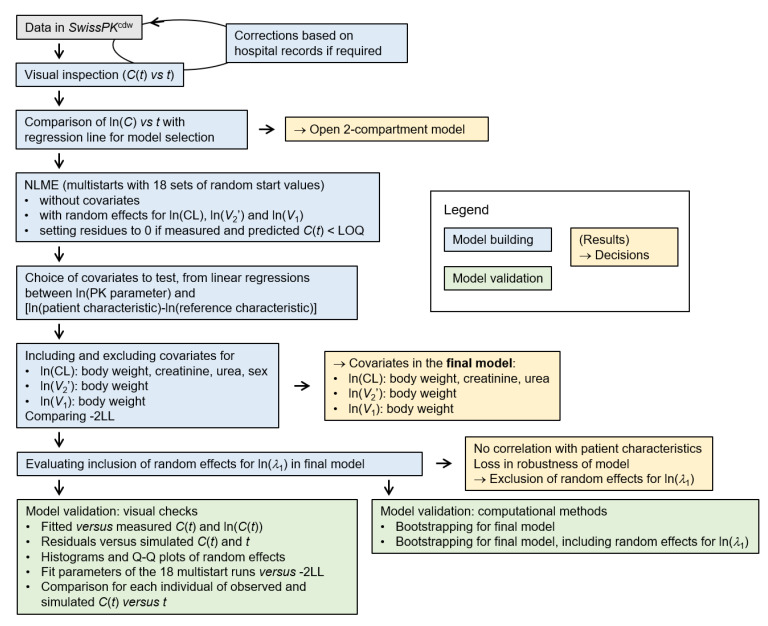
Workflow of the data analysis. Grey, data; blue, model building; yellow, decisions; green, model validation. See text for symbols and detailed explanations. Arrows, workflow.

**Figure 2 pharmaceutics-13-01596-f002:**
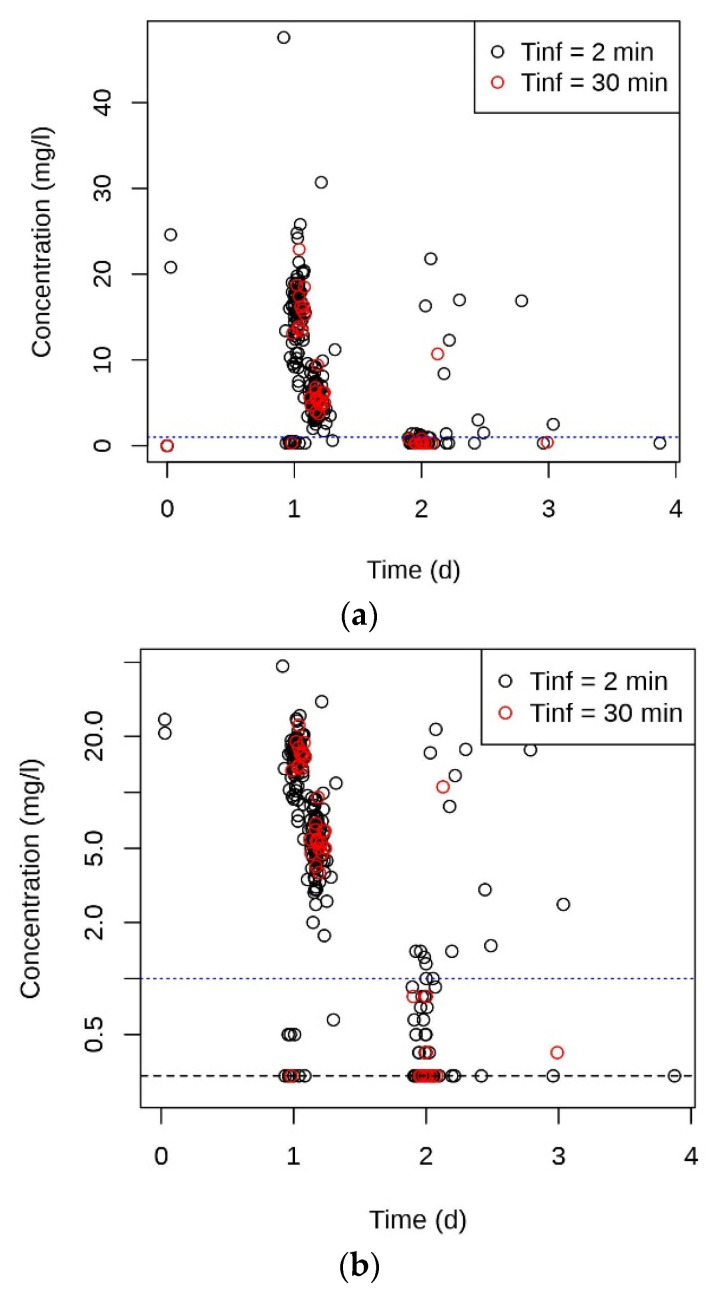
All measured gentamicin plasma concentrations of the 109 patients plotted against time after the start of the first of 2 or 3 infusions. (**a**) Linear, (**b**) logarithmic concentration scale. Most concentrations were determined during the 2nd dosing interval. Black and red symbols, concentrations after 2 and 30 min infusion, respectively. Blue dotted lines, recommended maximal trough concentration (1 mg/L). Horizontal broken black line in (**b**), LOQ (0.3 mg/L). All data were included for the modeling (up to 3 infusions). Concentrations < LOQ are indicated at LOQ in the Figure.

**Figure 3 pharmaceutics-13-01596-f003:**
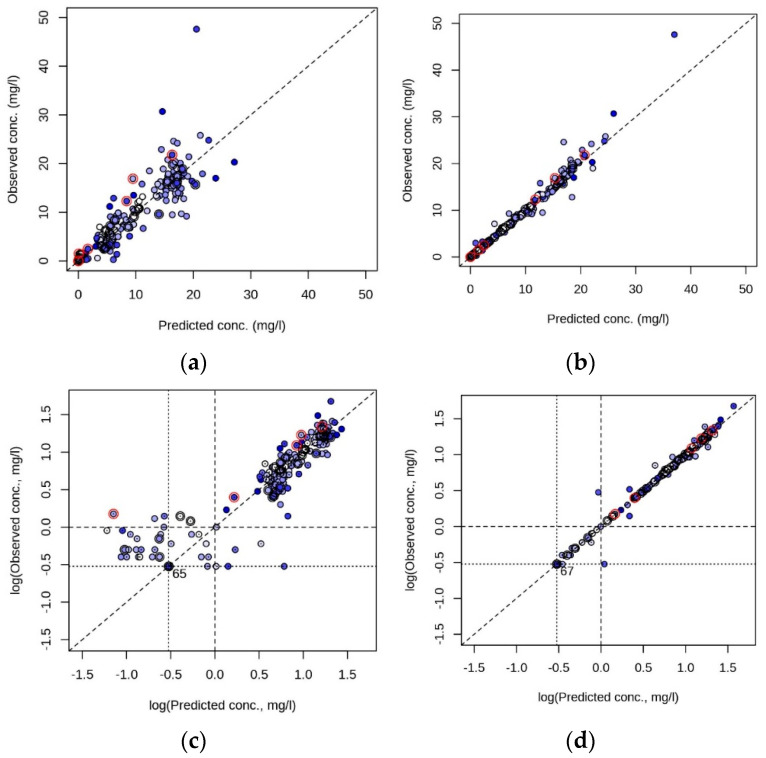
Comparison between measured (observed) and simulated (predicted) gentamicin plasma concentrations on the population (**a**,**c**) and individual (**b**,**d**) levels, in linear (**a**,**b**) and logarithmic (**c**,**d**) scales. Blue, light to dark indicates lowest to highest age. Black circle around symbol, gestational age at birth < 37 weeks. Red circle around symbol, serum creatinine > 50 µM. Diagonal line, line of unity. (**c**,**d**) Dot within a symbol, male; dotted lines, LOQ (0.3 mg/mL); broken lines, recommended maximal trough level (1 mg/L); number at LOQ intercept, number of data points at LOQ intercept (number of *C*(*t*) with measured and simulated value ≤ LOQ).

**Figure 4 pharmaceutics-13-01596-f004:**
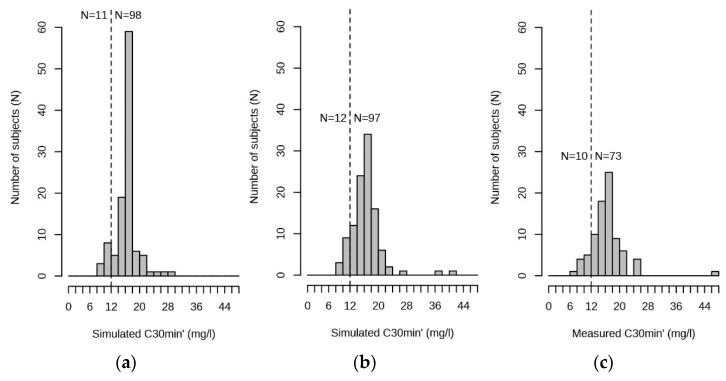
Distribution of the simulated and measured *C*(30 min’) values. (**a**) Simulated at population level; (**b**) simulated at individual level. (**c**) Measured concentrations within 1 h after infusion stops (available from 83 patients). Vertical broken line shows the recommended maximal *C*(30 min’) for long-duration treatment (12 mg/L). *N*, numbers of patients with *C*(30 min’) below 12 mg/L (left to the broken line), and above 12 mg/L (right to the broken line).

**Figure 5 pharmaceutics-13-01596-f005:**
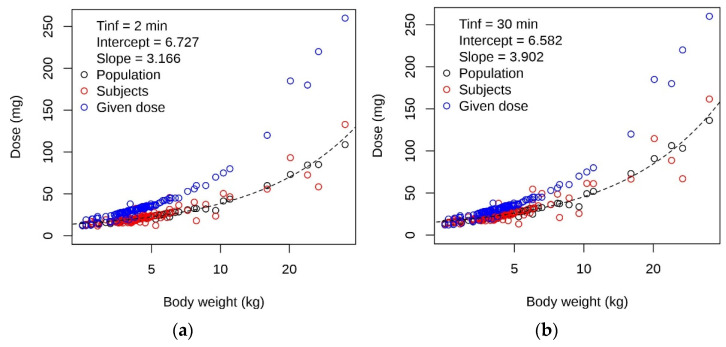
Predicted doses to reach *C*(30 min’) between 10 and 12 mg/L, calculated with the final model and compared to the given doses. (**a**) For 2 min and (**b**) for 30 min infusion duration. Predictions were made on the population (black symbols) and individual (red symbols) level. Blue symbols, given dose. Broken line, line with intercept (in mg) and slope (in mg/kg) from the linear regression analysis (predicted dose on individual level vs. body weight, parameters as indicated in the plots; note that bodyweight is in logarithmic scale).

**Figure 6 pharmaceutics-13-01596-f006:**
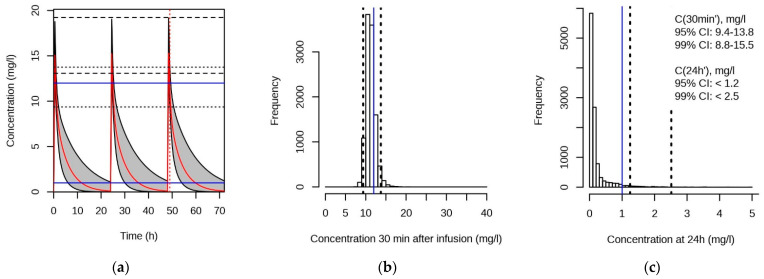
Plasma concentrations simulated with the final model to reach *C*(30 min’) between 10 and 12 mg/L. Simulations for 30 min infusion time. (**a**) Plasma-concentration time curves. Grey, 95% CI of the simulated concentrations (from 100 simulations per subject, i.e., 10,900 simulations in total). Red, median *C*(*t*) at each time point. Blue horizontal lines show recommended maximal trough (1 mg/L), and maximal *C*(30 min’) level for long-duration treatment (12 mg/L). Horizontal dotted and broken lines, 95% CI of the simulated *C*(30 min’) and maximal levels (at infusion stop), respectively. Vertical dotted red line, time point for *C*(30 min’). (**b**) Distribution of the simulated *C*(30 min’). Vertical blue line, 12 mg/L. Vertical black broken lines, 95% CI of the simulated *C*(30 min’). (**c**) Distribution of the simulated *C*(24 h’). Vertical blue line, 1 mg/L. Vertical black broken lines, 95% and 99% upper CI of the simulated *C*(24 h’). The 95% and 99% CI values for the simulated *C*(30 min’) and *C*(24 h’) are provided in panel (**c**).

**Figure 7 pharmaceutics-13-01596-f007:**
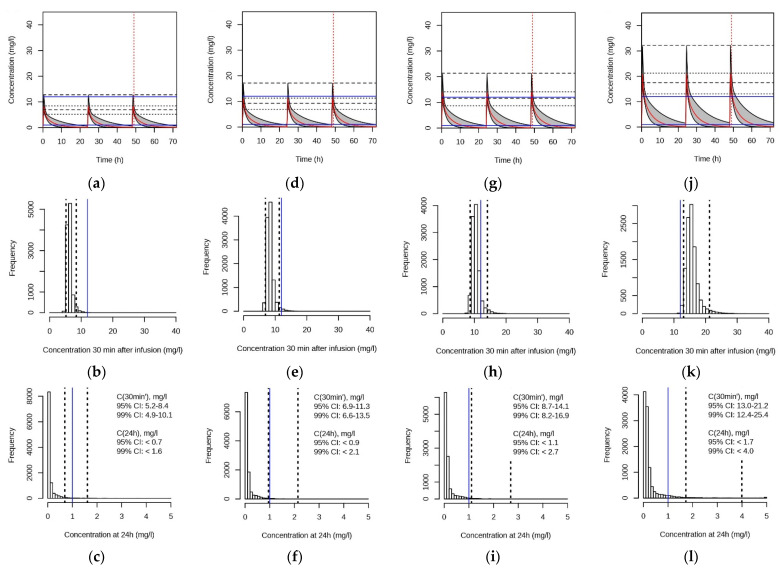
Simulations with the final model over 3 dosing intervals of 24 h each with 30 min infusion time. Once daily dose was (**a**–**c**) 3 mg/kg; (**d**–**f**) 4 mg/kg; (**g**–**i**) 5 mg/kg; (**j**–**l**) 7.5 mg/kg. (**a**,**d**,**g**,**j**) Simulated plasma-concentration time curves. (**b**,**e**,**h**,**k**) Respective distributions of simulated *C*(30 min’) of the third infusion. (**c**,**f**,**i**,**l**) Distributions of the simulated *C*(24 h’) of the third infusion. For further details, see Figure 6.

**Table 1 pharmaceutics-13-01596-t001:** Characteristics of the patient population, means (with *N*, number of subjects with available data), medians, ranges and standard deviations (SD).

Patient Characteristics	Mean	Median	Range	SD
Age (d)	213.1 (*N* = 109)	29.1	0.9–5299	750
Sex	67 male; 42 female			
Body weight (kg)	5.42 (*N* = 109)	4.23	2.5–35	4.64
Body surface area (m^2^)	0.293 (*N* = 109)	0.251	0.156–1.19	0.161
Height (cm)	58.7 (*N* = 109)	54.0	34–170	18.8
Gestational age (weeks) ^1^	39.0 (*N* = 94)	39.3	29–42	2.11
Serum creatinine (µM)	73 subjects with “< 27 µM” (*N* = 107)	n.a. ^2^	<27–72	n.a.
Serum urea (mM)	8 subjects with “< 1.8 mM” (*N* = 103)	3.20	<1.8–17.3	n.a.

^1^ The age of the patients with gestational age (GA) < 37 weeks (*N* = 11) was 19.5 d (GA 33/6 w/d), 2.1 d (GA 34/5 w/d), 1.0 d (GA 35/4 w/d), 13.3 d (GA 35/6 w/d) and older than 40 d; ^2^ n.a., not applicable.

**Table 2 pharmaceutics-13-01596-t002:** Indications with primary diagnoses and their frequencies in the study population.

Indications/Primary Diagnoses	Frequency ^1^
**Suspected bacterial infection at presentation**	**43 (8)**
Fever	34 (4)
Viral meningitis	6 (2)
Viral gastroenteritis	2 (2)
Hypovolemic shock	1
**Suspected superimposed bacterial infection**	**14 (1)**
Acute upper respiratory infection	10 (1)
Acute bronchiolitis due to respiratory syncytial virus	3
Influenza	1
**Proven bacterial infection**	**51 (15)**
Pyelonephritis	16 (3)
Central line-associated blood stream infection (CLABSI, confirmed or suspected)	8 (2)
Neonatal sepsis (confirmed or suspected)	9 (7)
Bacterial skin and soft tissue infection	3 (1)
Community-acquired Pneumonia	3
Ventilator-associated pneumonia	2 (1)
Hospital-acquired pneumonia	1 (1)
Sepsis	2
Mediastinitis	2
Bacterial meningitis	1
Multifocal arthritis	1
Bacterial parotitis	1
Endocarditis	1
Omphalitis	1
**Prophylaxis**	**1**
Surgical antimicrobial prophylaxis	1

^1^ In parenthesis, frequency among patients with 3 concentration measurements > LOQ (0.3 mg/L) within one dosing interval.

**Table 3 pharmaceutics-13-01596-t003:** Fit parameters with SE and *p* of the final model. Values in parentheses are results when setting concentrations below the LOQ to LOQ/2 (0.15 mg/L).

Parameter	Reference Value for Intercept	Fit Value	SE	*p*
Structural model				
ln(CL) = *θ*_1_ + *θ*_5_ × (ln(weight) − ln(4 kg)) + *θ*_6_ × (ln(creatinine) − ln(27 µM)) + *θ*_7_ × (ln(urea) − ln(3 mM)); CL = exp(*θ*_1_) × (weight/4 kg)^*θ*_5_^ × (creatinine/27 µM)^*θ*_6_^ × (urea/3 mM)^*θ*_7_^				
*θ*_1_, Intercept (ln(CL, l × d^−1^))		2.267 (2.232)	0.0375 (0.0325)	-
*θ*_5_, Δ ln(body weight, kg)	ln(4 kg)	1.219 (1.171)	0.0761 (0.0654)	<0.0001
*θ*_6_, Δ ln(serum creatinine, µM)	ln(27 µM)	−0.964 (−0.916)	0.203 (0.180)	<0.0001
*θ*_7_, Δ ln(serum urea, mM)	ln(3 mM)	−0.168 (−0.147)	0.0826 (0.0733)	0.019
ln(*V*_2′_) = *θ*_2_ + *θ*_8_ × (ln(weight) − ln(4 kg)); *V*_2_’ = exp(*θ*_2_) × (weight/4 kg)^*θ*_8_^				
*θ*_2_, Intercept (ln(*V*_2_’, l))		−0.174 (0.0937)	0.139 (0.0814)	-
*θ*_8_, Δ ln(body weight, kg)	ln(4 kg)	0.974 (1.314)	0.299 (0.111)	<0.001
ln(*λ*_1_) = *θ*_3 ;_ *λ*_1_ = exp(*θ*_3_)				
*θ*_3_, Intercept (ln(*λ*_1_, d^−1^))		3.644 (3.541)	0.201 (0.128)	-
ln(*V*_1′_) = *θ*_4_ + *θ*_9_ × (ln(weight) − ln(4 kg)); *V*_1_ = exp(*θ*_4_) × (weight/4 kg)^*θ*_9_^				
*θ*_4_, Intercept (ln(*V*_1_, l))		0.204 (0.205)	0.0779 (0.0572)	-
*θ*_9_, Δ ln(body weight, kg)	ln(4 kg)	0.688 (0.625)	0.0864 (0.0670)	<0.0001
Variance of the random effects (inter-individual variance)				
ln(CL)		0.107 (0.0830)	0.0972 (0.0125)	
ln(*V*_2′_)		0.291 (0.186)	0.0972 (0.0410)	
ln(*V*_1_)		0.0391 (0.0284)	0.0101 (0.0100)	
Residual error				
Residual error		0.102 (0.141)	0.0035 (0.0051)	

**Table 4 pharmaceutics-13-01596-t004:** Ranges of PK parameters for the final model and for the studied patient population. Population level (individual level).

Parameter	Minimal Value	Mean Value	Median Value	Maximal Value
CL (mL/min)	2.76 (1.47)	9.39 (9.84)	6.88 (6.84)	73.2 (71.0)
CL (mL/min per 70 kg)	42.4 (19.7)	114 (121)	117 (118)	177 (298)
CL (mL/min per 1.73 m^2^)	21.8 (8.7)	49.4 (52.3)	48.4 (48.0)	113 (138)
*V_z_* (l) (*V*_1_ + *V*_2′_)	1.42 (0.915)	2.58 (2.71)	2.16 (2.24)	12.4 (16.0)
*V_z_* (l per 70 kg)	24.8 (14.9)	35.3 (37.1)	35.8 (36.5)	39.7 (104)
*t*_1/2_ (h) (ln(2) × *V*_z_/CL_z_)	1.72 (1.22)	3.79 (4.06)	3.53 (3.51)	8.38 (16.9)
*λ*_1_ (h^−1^)	1.59 (1.59)			
*t*_1/2_ of distribution phase (h) (ln(2)/*λ*_1_)	0.435 (0.435)			
*V*_1_ (L)	0.887 (0.762)	1.45 (1.46)	1.27 (1.27)	5.46 (7.06)
*V*_1_ (L per 70 kg)	10.9 (9.50)	20.6 (20.6)	21.1 (20.5)	24.8 (33.4)

## Data Availability

Secure access to the data presented in this study is available on request according to the data sharing policies of SPHN and *SwissPK*^cdw^ [16]. The data are not publicly available due to confidentiality reasons (Swiss Human Research Act, 810.30).

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
