# Peer review of "Gentamicin Population Pharmacokinetics in Pediatric Patients—A Prospective Study with Data Analysis Using the saemix Package in R"

_pharmaceutics, 2021, doi:10.3390/pharmaceutics13101596_

Round 1
Reviewer 1 Report
In this manuscript, the authors described population PK modeling for selecting the recommended dosing in pediatric patients. Since gentamicin is a drug that requires TDM, population modeling was performed in pediatric patients using PK data obtained from sparse blood sampling. The knowledge of gentamicin PK in pediatric patients is an important topic, especially for a future pediatric dose selection of gentamicin. The comprehensive finding presented in the manuscript is interesting and useful for the research community. I would suggest some points in the manuscript to improve to be accepted for publication in Pharmaceutics.
- Line 162-163: What are the “defined physiologically meaningful ranges for the fit parameters”? Please provide the physiologically meaningful range values of each parameter and add references.
- The author lacked a description of the validation of the final model. Please add the model validation to the methods and results section and elaborate more.
Reviewer 2 Report
In this study, the author analyzed the Gentamicin population pharmacokinetics by using the non-linear mixed-effects modeling approach and predicted the dosing scheme. Major Reviews Pros:- The concept and framework of the study design are great and very comprehensive
- Use origin PK data in the modeling process
- Provide the source code in the public repository
- The workflow is not easy to follow.
- Cannot reproduce the result from the provided source code due to the data confidential issue.
- I would like to recommend adding a workflow diagram to help readers easy to understand the modeling process.
- The study used "whole" data in model calibration rather than separate the data into "calibration group" (using 2/3 PK data to find the parameter value) and "validation group" (using 1/3 PK data to exam the model performance). Why didn't use this common approach in this study?
- After read through the whole manuscript, I still cannot fully understand the definition of the population and subject level. Does it mean Inter-individual (population) and intra-individual (subjects)? Also, I can't understand why there are two calibration results (population and subject) in Figure 2. Does there any clear definition or source code that can help to understand the setting that used in this study?
- L158-163. This study used parallel computation with multi-starts. How is the converge result for these simulations? How many sample numbers that required to reach the convergence?
- L197. I want to understand the definition of mean R_ac, is it the population mean with its distribution? Or, it is R_ac with mean and sd (or se).
- L250. Why does the author determine that lambda_1 is the parameter that has an overfitting issue? Did you conduct any examination to conclude that?
- Figure 4. Why do the blue circles do not have the same trend as red and black circles?
- Figure 5. How many total numbers of simulations that generated in this result? 109 subject x 100 simulations?
- I would like to ask if the author can release some part of the data (insensitive data) in the df5.csv file mentioned in the GitLab repo, which can help the reader to conduct reproducible research.
- Figure S3. "The general underestimation ... rather than one-compartment model". What does this mean?
- Figure S11, S12. For my understanding. This is the plot that compares the observation and prediction under the individual (subject) level to exam the model performance. Therefore, why add the prediction of population-level in this plot? What information do you want to reveal?
Reviewer 3 Report
Dear Authors,
thank you for the study.
I have a few suggestions that can help make presentation clearer.
The sentence "All patients aged 1 d to 10 y receiving gentamicin for at least 48 h at the University Children’s Hospital Zurich (Switzerland) between ..."
seemed to be duplicated in "Materials and Methods" section and in "Results" section.
I suggest to leave information about the number of patiens in the study in Materials and Method section.
2. Could you please provide the protocol of the study as a part of Supplementary Materials. (all primary disorders, exclusion and inclusion criteria etc.)?
3. Please include the description of comorbidities to the primary diagnoses (Table 2). It looks strange that primary diagnos is Influenza and gentamicin is prescribed. It definitely needs to be more detailed.
With best regards.
Round 2
Reviewer 2 Report
I appreciate the hard work from the authors. The manuscript is good to publish under the current format.